# Cell-Based Treatment in Acute Myeloid Leukemia Relapsed after Allogeneic Stem Cell Transplantation

**DOI:** 10.3390/biomedicines12081721

**Published:** 2024-08-01

**Authors:** Martina Canichella, Paolo de Fabritiis

**Affiliations:** 1Hematology Unit, St. Eugenio Hospital, ASL Roma2, 00144 Rome, Italy; paolo.defabritiis@aslroma2.it; 2Department of Biomedicine and Prevention, Tor Vergata University, 00133 Rome, Italy

**Keywords:** acute myeloid leukemia (AML), allogeneic stem cell transplantation (ASCT), cellular therapy, donor lymphocyte infusion (DLI), chimeric antigen receptor (CAR)-T cells, next-generation CAR-T cells, universal CAR-T (UCAR-T), UniCAR technology

## Abstract

Allogeneic stem cell transplant (ASCT) remains the only treatment option for patients with high-risk acute myeloid leukemia (AML). Recurrence of leukemic cells after ASCT represents a dramatic event associated with a dismal outcome, with a 2-year survival rate of around 20%. Adoptive cell therapy (ACT) is a form of cell-based strategy that has emerged as an effective therapy to treat and prevent post-ASCT recurrence. Lymphocytes are the principal cells used in this therapy and can be derived from a hematopoietic stem cell donor, the patient themselves, or healthy donors, after being engineered to express the chimeric antigen receptor (CAR-T and UniCAR-T). In this review, we discuss recent advances in the established strategy of donor lymphocyte infusion (DLI) and the progress and challenges of CAR-T cells.

## 1. Introduction

To date, acute myeloid leukemia (AML) represents the most frequent indication for allogeneic stem cell transplantation (ASCT). According to the European Leukemia Network (ELN) 2022, the European Society for Blood and Marrow Transplantation (EBMT), and American Society of Blood and Marrow Transplantation (ASBMT), ASCT is indicated in eligible high-risk AML patients (whether due to cytogenetic, molecular, or measurable residual disease (MRD) factors) with an age ≤75 years. Relapse post-ASCT remains an unmet need with low rate of survival (2-year survival rate of 20%) [1,2,3,4,5,6]. In this context, we currently have three major strategies: targeted therapy for cases with a genetic mutation, hypomethylating agent based-therapy, and cellular therapy. The purpose of this review is to illustrate the progress of cellular therapy in AML that has relapsed post-ASCT. This strategy represents a treatment potentially able to reverse the poor prognosis of these patients. The pioneer of adoptive cellular therapy is donor lymphocyte infusion (DLI), but new generations of CAR-T cells are also emerging (Figure 1). The introduction of these cellular strategies into clinical practice will lead to planning post-ASCT strategies for high-risk patients with a “prophylactic” approach, in order to prevent frank relapse.

## 2. Donor Lymphocyte Infusions (DLI)

Based on the premise that the graft-versus-leukemia (GvL) effect constitutes the immunological backbone of ASCT, DLI has always been an attractive strategy to reinforce antitumor activity to finally reduce the relapse rate post-ASCT. However, based on the overlapping graft-versus-host effect caused by DLI, different studies have investigated the optimal time of DLI administration considering three possible scenarios post-ASCT: prophylaxis, if no signs of disease can be detected (pro-DLI); pre-emptive, if MRD and/or mixed chimerism (pre-DLI) is present; and therapeutic DLI in case of frank hematological relapse. DLI has been proven to have maximum effectiveness in the first two contexts, while, in hematological relapse, activity is limited. Furthermore, a second field of investigation has been the optimal dose of DLI able to minimize the GvHD while maintaining robust GvL activity. Donor T cells can be separated and cryopreserved from the initial peripheral blood mobilization of hematopoietic stem cells (HSC) or obtained later by an unstimulated leukapheresis from the original donor. The donor lymphocytes commonly administered in clinical practice are unmanipulated and composed of different CD3+ T cell subsets, mostly T cells, which recognize host antigens through presentation by major histocompatibility complex (MHC) class I or II [7].

### 2.1. DLI, the Past

The first successful use of DLI was in 1990, in three patients with chronic myelogenous leukemia (CML) [8]. This study represented the milestone of post-ASCT immunotherapy, demonstrating the feasibility of this approach, especially in the context of cytogenetic/molecular relapse. At the same time, several factors emerged that influenced the response, such as a longer remission period after ASCT and disease control at the time of DLI [9]. Subsequently, several studies have explored the optimal dose and frequency of infusions to limit the onset of GvHD [10]. After the first report, in which DLI was administered at the dose of 4 × 10^8^ cells/kilogram (kg) of body weight resulting in the inevitable development of GvHD, the dose commonly accepted was ≤1 × 10^7^ T-lymphocytes/kg. Indeed, the EBMT guidelines recommended a dose of 1 × 10^5^/kg or 1 × 10^6^/kg for prophylactic and pre-emptive contexts [11]. Currently, most evidence regarding DLI in clinical practice derives from retrospective series in which the results demonstrated considerable heterogeneity, depending on the state of the disease, the dosage of DLI, and the duration of remission. Taken together, OS ranged from 25 to 80% while the incidence of relapse ranged from 20 to 43%. Overall, the onset of acute GvHD was relatively low (8–30%), but the chronic GvHD increased between 30 and 55%. In particular, the efficacy of DLI was limited when applied in hematological relapse [11,12]. A large retrospective study from the EBMT group reported OS of 21% for patients receiving DLI, compared to 9% for those who did not receive treatment. Low tumor burden was one of the factors associated with successful treatment in multivariate analysis [13].

### 2.2. DLI, the Present

Given the limited efficacy of therapeutic DLI, research proposed a pre-emptive approach, the less proliferative stage of MRD positivity and/or increasing mixed chimerism where DLI is associated with high response rate and better outcome. One of the first experiences was reported by Dominietto et al., who investigated the role of MRD as predictor of relapse and the protective role of DLI infusion in delaying the relapse itself [14]. The study showed that the patients who were MRD-positive who did not receive DLI presented a higher relapse rate compared with patients who were MRD-positive who received DLI. Indeed, two prospective trials demonstrated the preventive benefit of pre-DLI in reducing relapse rate [15,16]. Taken together, the pre-emptive use of DLI in patients with MRD or mixed chimerism can reduce relapse and improve survival. Similarly, pre-DLI administration based on mixed chimerism resulted in complete chimerism conversion, low relapse rate, and better OS compared with patients who did not receive DLI. The frequency of DLI-induced GvHD varies by reports and can be influenced by T cell numbers, number of cycles, and the immunosuppression state of the patients. Overall, the incidence of chronic GvHD is around 40% of cases.

### 2.3. DLI, the Future

A growing body of evidence based on a retrospective series suggests that the optimal use of DLI administration is in prophylaxis for high-risk AML and myelodysplastic syndrome (MDS). Generally, prophylaxis can be considered a form of maintenance therapy in which the drug, in this case the cellular product, is administered after achieving the deepest possible complete remission. Schmid et al. reported a matched-pair analysis on 89 patients and 89 control cases with similar characteristics, demonstrating that pro-DLI led to better OS in a very high-risk subgroup [17]. Indeed, Legrand et al. showed that patients who received pro-DLI had a 2-year overall survival (OS) rate of 75% and a relapse rate of 22% [18]. Similarly, Jedickova et al. reported a lower relapse rate (22% vs. 53%) and a better OS (67% vs. 31%) for the pro-DLI group, compared with those who did not receive DLI [19]. Recently, increased interest has been shown in the use of DLI in the context of haploidentical-ASCT (haplo-ASCT). In a retrospective study by Couchois et al., 36 patients were treated with DLI, after haplo-ASCT using post-transplant cyclophosphamide (PTCy) as GvHD prophylaxis [20]. Twenty-five of these patients had a diagnosis of AML/MDS with a 1-year overall survival (OS) and progression-free survival (PFS) of 83% and 76%, respectively. The incidence of GvHD was 33%, which was not different from that observable in patients who receive transplants from different types of donors. Indeed, Santoro et al. reported a multicenter study which compared the outcome of DLI used in prophylaxis, preemptive and relapse: using the prophylactic approach, OS proved to be superior, but at the same time the incidence of cGvHD was higher [21]. Currently, a prospective randomized study (ELIT trial NCT00674427) is ongoing which will determine the incidence and severity of acute and chronic GvHD associated with repetitive dosing of DLI.

Overall, the current EBMT recommendations for the use of DLI are as follows:The first infusion is recommended after cessation of immunosuppression for >30 days and is not recommended with active GvHD and infections;The median interval from ASCT to first DLI across all studies was 4–6 months, while an early application could be desirable in high-risk diseases;The infusions should have a dose increase of 0.5–1 log with intervals of 4–6 weeks;DLI frequencies should be guided by response (MRD, degree of chimerism) and the occurrence of GvHD has to be considered as limiting toxicity [11].

### 2.4. Beyond Unmanipulated DLI

To enhance the efficacy and reduce the side effects of DLI, various strategies have been developed. These strategies aim to increase the therapeutic potential while reducing adverse effects. These approaches include the following:-Activation of cell subsets by cytokines or growth factor;-Selection and expansion of specific T cell subsets;-Infusion of natural killer (NK) cells.

Although these strategies are effective and promising when applied in the context of MRD, they have only been used in the context of clinical trials thus far.

#### 2.4.1. Cytokine-Induced Killer (CIK) Cells

CIK consists of donor peripheral blood mononuclear cells obtained after in vitro stimulation with anti-CD3, IFN-γ, and recombinant human IL-2. As described for the first time by Schmidt-Wolf et al., this stimulation induces expansion of T cells and CD56 expression on CD3/CD8 T lymphocytes [22]. CIK cells acquire “double” T cell capacity of both T and NK cells and can exert a potent, MHC-unrestricted antitumor activity via the NKG2D receptor [23]. Introna et al. studied the sequential administration of DLI and CIK cells in 73 patients and reported encouraging results in terms of disease control, particularly with MRD positivity [24]. Indeed, a retrospective analysis where DLI and CIK were compared in 91 patients (DLI, n = 55; CIK, n = 36), the CIK approach showed a better outcome. The 6-month OS and cumulative incidence of relapse were superior for the CIK group (77% vs. 57%, and 22% vs. 55%, respectively). Moreover, patients treated in overt hematological relapse showed the poorest outcome, as expected [25].

#### 2.4.2. G-CSF-Stimulated DLI

Because DLI can be administered only after a minimum of 3–4 months after relapse and the early AML relapse post-ASCT showed the poorest outcome, several studies have addressed the possibility of using G-CSF-mobilized peripheral blood mononuclear cells to produce DLI due to the immune regulatory effects of G-CSF [26]. The increased presentation of the antigen, therefore, could be the rationale for allowing an earlier application of DLI, even during immunosuppression. Several studies have demonstrated the feasibility and efficacy of this approach in high-risk AML. Similar results were obtained when G-CSF-stimulated DLI was administered after mild chemotherapy [27].

#### 2.4.3. Antigen-Specific DLI

A further approach to increasing αβ T cell specificity is the generation of more malignancy-specific donor cells against polymorphic MiHA. This strategy includes many drawbacks represented by technical difficulties. However, αβT cell subsets have been developed that are directed against leukemia-associated antigens (LAA), for example, WT1, MAGE, or BCMA [28]. Another ideal target for individualized immunotherapeutic approaches is NPM1. Recently, Greiner et al. described specific T cell responses of CD4^+^ and CD8^+^ T cells against epitopes derived from mutated regions of NPM1 in 25 patients with the NPM1mut characteristic. Despite the small number of patients, this study showed promising results which might be translated into maintenance treatment or apersistent MRD context [29].

#### 2.4.4. DLI in Combination with Other Drugs

Experiments with azacytidine (AZA) plus DLI administered as maintenance post-ASCT have been limited to retrospective series. Guillame et al. reported the outcomes of 77 patients with high-risk MDS and AML treated with this strategy both in the prophylactic and preemptive phases. OS and PFS at 24 months were 70.8% and 68.3%, respectively. The cumulative incidences of grade II-IV acute GvHD and chronic GvHD were 27.4% and 45%, respectively. Despite the retrospective nature of the study, these data support the hypothesis that maintenance treatment with AZA and DLI is a valid option with acceptable toxicity mostly in high-risk patients [30]. In pediatric settings, a recent study reported a better OS and leukemia-free survival in children who received maintenance AZA plus DLI compared with historical controls [31].

## 3. CAR-T

In contrast to the success achieved in the field of specific B-cell lymphoproliferative malignancies, the clinical application of CAR-T cells in AML has shown considerable potential, although it remains an area of ongoing research and refinement [32,33,34,35,36,37,38]. CAR-T cells are derived from T cells of either patients or healthy donors (universal CAR-T, see below), engineered to express CAR on their surface. Unlike the T cell receptor (TCR), CAR enables the recognition of antigens present on cancer cells, independently of MHC molecules, thus preventing cancer cells from evading immune system surveillance due to reduced MHC expression on their surface. The CAR structure comprises four regions with specific functions: the single-chain variable fragment (scFv) is the extracellular binding domain; the hinge is responsible for flexibility and alignment with the target antigen; the transmembrane domain; and finally the intracellular signaling domain [39]. Currently, five generations of CARs are available, differing mainly in the structure of the intracellular domain [40].

The first generation of CAR-T cells included constructs with CD3ζ but lacked co-stimulatory domains. These CAR-T cells demonstrated poor persistence and activity in vivo. The second generation introduced additional co-stimulatory domains, CD28 and 4-1BB, which significantly improved the efficacy and persistence of CAR-T cells and are currently approved for clinical use. To enhance safety, a third generation was developed, incorporating multiple co-stimulatory domains. However, this did not result in a significant improvement over the second generation. The fourth-generation CAR-T cells, known as T cells redirected for universal cytokine-mediated killing (TRUCKs), included an added IL-12, which can be either expressed constitutively or inducible after CAR activation. This enhancement supports the elimination of malignant cells through cytokine-mediated mechanisms. Finally, the fifth generation of CAR-T cells were designed to include the IL-2 receptor in their structure, aiming to further improve their therapeutic potential. In AML, however, the major limitation in the effectiveness of CAR-T is the absence of an antigen that meets the optimal criteria to be a CAR target, that is, specificity, stability, and the absence of normal HSC and other tissue to avoid off-target effects.

The success of CAR-T cell therapy also depends on the management of specific toxicities and on the strategy to overcome CAR-T limitations. The most common side effects of CAR-T are represented by cytokine release syndrome (CRS) and immune-effector-cell-associated neurotoxicity syndrome (ICANS) [41,42]. Indeed, in the case of CAR-T in AML, in which the target antigens are expressed both on blast and on HSC, myelosuppression has been observed to be an important limitation. For this reason, different strategies have been developed, which will be discussed in the following sections.

The current application context for CAR-T therapy is in primary refractory patients or those with post-transplant relapse, in both cases within clinical trials. In this section, we report the latest preclinical and clinical results regarding the most promising CAR-T cells targeted against CD33, CD123, CLL-1, and FLT3 antigens (Table 1).

### 3.1. CD33-CAR-T

CD33 belongs to the sialic-acid-binding immunoglobulin-like lectin (Siglec) transmembrane receptor family [43]. CD33 is a suitable target for immunotherapy, given it is expressed in about 90% of AML leukemic cells (both stem cells and blasts). Immunotherapy based on anti-CD33 constructs has been explored for years and Gentuzumab Ozogamicin (Mylotarg^®^), a monoclonal antibody conjugated with calicheamicin against CD33, is the proof of effectiveness of this target approach [44]. CD33-CAR-T showed great effectiveness against leukemic cells in the first preclinical studies with two major limitations: the functional failure of CAR-T cells and myelosuppression [45]. To overcome these side effects, the researchers investigated different constructs. Kenderian et al. developed a CD33-CAR-T with transiently expressed modified RNA, resulting in similar response to lentivirally transduced CAR-T with better response and prolonged survival [46]. Indeed, Li et al. and Qin et al. explored CAR-T with different co-stimulatory and antigen domains [47,48]. The results showed that CD33-CAR-T with the 4-1BB domain increased the central memory compartment while CAR-T with lintuzumab as the CD33 binding domain resulted in higher inhibition of leukemia proliferation. Moreover, the novel technological strategy (see below) of Universal CARs (UniCARs) has been employed in this context with the aim of reducing the myelosuppression off-target effect [49].

### 3.2. CD123-CAR-T

CD123 corresponds to the alpha chain of the human interleukin-3 receptor (IL-3Rα), which is frequently overexpressed in several hematological malignancies, including AML, but also on normal hematopoietic stem cells (HSCs) [50]. The first preclinical studies showed that the efficacy of CD123-CAR-T was associated with significant myeloablation. Despite its efficacy, the main limitations of CD123-CAR-T resulted from myelosuppression and immune escape by blast cells. To overcome these issues, Tasian et al. focused their efforts on blocking the action of CAR-T responsible for aplasia, using three different strategies: (i) a transient anti-CD123 matrix RNA through electroporation of CAR-T cells (RNA-CART123); (ii) the use of alemtuzumab to eliminate CAR-T; and (iii) the administration of rituximab in dual CAR-T (CART123-CD20) [51]. The immune-escape of blasts from CAR-T surveillance could be overcome by DUAL-CAR cells which co-express two CARs against different antigens or a bispecific CAR (TanCAR-T cell) in which two binding domains targeting two different antigens are expressed on the same CAR. In a mouse xenograft model, Petrov et al. investigated the effectiveness of anti-CD123-CD33 Dual CAR-T and demonstrated the elimination of all blasts [52].

Another approach to limit off-target effects such as myelotoxicity has been the development of Reversed CARs (RevCARs) (see Figure 2). RevCAR, like UniCAR, is composed of an effector module (EM) and a tumor-specific target module (TM). In RevCAR, the peptide epitope is an EM element, while the peptide epitope-binding domain is one of the TM elements [53]. Kittel-Boselli et al. developed a RevCAR-T with two different TMs, CD123 and CD33, able to target CD123+ or CD33+ on the blast cells. The construct was shown to be very efficient in eliminating tumor cells. Subsequent in vitro studies confirmed these results [54].

Moreover, UniCAR-T anti-CD123 have also been explored in this context, and in the first application UniCAR-T showed high effectiveness and low hematotoxicity against HSCs. The most promising results with the use of the CD123 target antigen came from the allogeneic CAR-T developed with a platform of gene-editing to eliminate the TCR α/β responsible for GvHD (see below); however, clinical results are not yet conclusive. In 2019, Sun et al. reported results obtained with second-generation anti-CD123 UniCAR-T cells. The patient was heavily pretreated and, after anti-CD123 CAR-T, achieved complete remission. Indeed, the clinical trial NCT04230265 is investigating autologous UniCAR02-T cells with TM123.

### 3.3. CLL1-CAR-T

C-type lectin-like molecule-1 (CLL-1) is a type II transmembrane glycoprotein expressed in more than 90% of AML blasts and leukemia stem cells (LSCs) and absent in normal CD34+CD38-HSCs [55,56]. This pattern of expression makes it a potential target for therapeutic intervention. The first experiment with the use of CLL1-CAR-T referred to Tashiro et al., who used a construct with 4-1BB as co-stimulatory domain. These cells showed significant cytotoxicity against several leukemic cell lines; however, in the mouse model, high toxicity was observed. The next step was the introduction of an inducible caspase 9 (iC9) to block the CLL1-CAR-T activity or anti-TNFα antibodies to reduce the CRS. Lin et al. explored anti-CLL-1 CAR-T cells in combination with programmed death receptor 1 (PD-1) silencing, demonstrating a higher efficacy in T cell exhaustion [57]. Subsequently, Zhang et al. reported the first clinical study on CLL1-CAR-T in a young 10-year-old patient with secondary AML. The patient experienced CRS grades 1–2 and at the end of treatment achieved CR with MRD negativity [58]. Several other groups have reported the efficacy and safety of CLL1-CAR-T, developing different constructs. Ma et al. reported the use of anti-CLL-1 CAR-T cells with PD-1 knockout in two patients who achieved CR and MRD negativity, and Pei et al. published the treatment with different costimulatory domains CLL1-CAR-T in seven children [58,59]. The results showed that the overall response rate (ORR) for patients with CD28/CD27 CAR-T cells was 75%, and for those with 4-1BB CAR-T cells it was 67%, while 57.1% was the 1-year survival rate [59].

### 3.4. FLT3-CAR-T

The FMS-like tyrosine kinase (FLT3) belongs to the class III receptor tyrosine kinase family [60]. FLT3 is fundamental to normal hematopoiesis, supporting cell survival, proliferation, and differentiation. It is expressed on HSCs but is also present on AML leukemic cells at a rate of 54–92% [61]. FLT3 gene mutations, located on chromosome 13, are the most frequent AML molecular alteration. The internal tandem duplications (ITD) occur in 15–30% of patients while mutations in the tyrosine kinase domain (TKD) occur in 5–10% of patients. These mutations result in constitutive activation of the FLT3 receptor, leading to increased survival and proliferation of blast cells. The prognosis for these cases, especially for ITD mutation, is very poor, resulting in event-free survival and OS rates of 12% and 16.6%, respectively. Chen et al. evaluated the FLT3 CAR-T in a preclinical model, demonstrating significant cytotoxicity due to increased production of IFN-γ [62]. Other study groups developed FLT3-CAR-T targeting its natural ligand. Wang et al. created second-generation CAR-T cells with a 4-1BB costimulatory domain and experimented with the FLT3 ligand (FLT3L)-binding domain as the antigen-binding domain [63]. The authors found that FLT3L CAR-T cells showed significant cytotoxicity and elimination of FLT3-ITD-mutated leukemic cells, while wild-type FLT3 cells were able to survive. Based on these results, it can be concluded that FLT3L-CAR-T cells, therefore, could emerge as an effective construct to treat AML with the FLT3-ITD mutation. Indeed, a further step forward was to combine FLT3L-CAR-T with other treatments. Jetani et al. explored the combination of anti-FLT3 CAR-T cell therapy with the FLT3 inhibitor crenolanib [64], resulting in higher efficacy in vitro and in vivo. Similarly, Li et al. experimented with dual-FLT3scFv/NKG2D-CAR-T cells plus gilteritinib [65], which showed the same cytotoxicity both on blasts and on HSCs. To reduce the risk of myelotoxicity associated with anti-FLT3 CAR-T cell therapy, Sommer et al. incorporated a safety switch into the CAR construct [66]. The authors confirmed that co-culturing anti-FLT3 CAR-T cells led to a significant reduction in the number of HSCs. To enhance the safety of this therapy, they added two mimotopes of rituximab (R2 off-switch) to the CAR construct between the hinge region and the scFv. The modified cells (anti-FLT3 CAR-R2 T) demonstrated no reduction in efficiency compared to CAR-T cells without an off-switch. Further studies validated the effectiveness of anti-FLT3 CAR-R2 T cells in eradicating leukemia, and the administration of rituximab resulted in the depletion of CAR-T cells, thereby limiting hematotoxicity and allowing for bone marrow recovery.

## 4. Novel Strategies for CAR-T in AML

Continuing efforts to enhance the specificity and efficacy of CAR-T while mitigating off-tumor toxicities have been expanded into AML. In the following section, we report the more promising approaches in preclinical studies to overcoming some of these drawbacks.

## 5. Strategies to Improve CAR-T Specificity

To increase the potency of CAR-T cells in AML, two main approaches have been studied: the upregulation of the antigen on blast cells and the modification of the CAR receptor. Based on the evidence that azacitidine increases the expression of CD123, El Khawanky et al. tested a third-generation CD123-CAR against MOLM-13 AML cell lines, resulting in a low tumor burden. Another compound able to increase different AML antigens is ATRA and FLT3 inhibitors [67]. The optimization of the CAR design is also crucial for CAR-T therapy success. In this field, both the scFv and the hinge domain can be modified to increase the avidity with the ligand. Pérez-Amill et al. changed the order of the heavy and light chain domains in the scFv and the linker length, which showed a higher efficacy against AML cell lines [68]. Finally, Mandal et al. presented a method based on cross-linking mass spectrometry and emphasized the importance of determining the structural conformation of the antigen on target [69].

## 6. Dual CAR-T

The mechanism of antigen escape allows tumor cells to evade the immune response, posing a significant challenge in cancer treatment. Recent studies have shown that in approximately 30–70% of B-cell hematological malignancies, the CD19 antigen is downregulated or completely lost after relapse. To address this limitation, researchers have developed dual CAR-T cells. These advanced CAR-Ts are designed to recognize and target two different antigens simultaneously, thereby increasing the likelihood of targeting tumors. Ghamari et al. have developed a tandem CAR targeting both CD123 and folate receptor β, which are upregulated on blasts and leukemic stem cells (LSCs) from patients with AML [70]. This tandem CAR has demonstrated the ability to secrete higher levels of interleukins, particularly IFN-γ and IL-2, compared to single CARs. The increased secretion of these cytokines results in enhanced T cell activation and a higher rate of tumor cell lysis, thereby improving the functionality and efficacy of tandem CARs in comparison to single CAR. Indeed, Scherer et al. described a novel ligand-based CAR targeting CD70 in combination with either CD123 or CLL-1 [71]. This CAR incorporates the CD27 scFv, which serves as the natural ligand for CD70. The combination of this ligand CD70 CAR with CD123 or CLL-1 demonstrated enhanced antitumor efficacy both in vitro and in murine models, outperforming single CAR even against tumors with low antigen density. However, a significant limitation of dual CAR-T is the elimination of cells expressing only one of the target antigens, which include normal cells. To address this issue, Haubner et al. developed a novel strategy called IF-BETTER gate to reduce on-target off-tumor toxicity and counteract antigen escape in heterogeneous AML blasts. The IF-BETTER gated CAR-T cells approach involves co-targeting two antigens—one with distinct expression patterns on AML cells and HSCs, and a second antigen exclusively expressed on AML cells—thus minimizing on-target off-tumor toxicities [72]. The first antigen is ADGRE2, highly expressed on AML cells (>1.0 × 10^3^ molecules per cell) and minimally expressed on normal HSCs (<1.0 × 10^3^ molecules per cell). The second target is CLEC12A, which is co-expressed with ADGRE2 on LSCs but not on HSCs. The authors developed a sensitivity-tuned ADGRE2-CAR with a CLEC12A-targeted chimeric costimulatory receptor, demonstrating enhanced activity against LSCs without increasing toxicity to HSCs in vitro and in murine models. A phase 1 clinical trial has been initiated to evaluate this strategy in patients with R/R AML (NCT05748197).

## 7. Strategies to Reduce Toxicity: The Gene Editing Approach

The gene editing strategy represents a promising approach to reducing CAR-T toxicity and enhancing efficacy. The applications of this technology vary significantly. One of the most interesting consists of using CRISPR/Cas9 to delete CAR-T targets such as CLL-1 or CD33 from HSCs before transplantation [73]. This approach aims to prevent off-tumor effects associated with CAR-T therapy and limit the cytotoxic effect of CAR-T cells [74]. Editing approaches consist of non-essential antigen targeting; however, they come with two significant limitations. Firstly, the pool of truly dispensable targets is likely to be limited. Secondly, target functional redundancy might enable escape mechanisms through marker loss or under-expression, without adversely affecting leukemic cell fitness. Wellhausen et al. have proposed a novel approach involving the editing of the pan-hematological epitope CD45 from both HSCs and CAR-T cells through CRISPR base editing. The edited anti-CD45-CAR-T cells did not recognize HSC but did recognize CD45+ AML blasts. At the same time, the edited CD45 on the CAR-T cell prevented fratricide into the CD45 epitope. This allows CAR-T cells to evade HSC recognition, preserving the essential function of CD45. The field of gene editing has another highly significant application in the platform of allogeneic CAR-T cells, universal CAR-T (UCART), or “off the shelf” CAR-T (see below).

## 8. UCAR-T

To overcome limitations regarding the manufacturing process of autologous CAR-T, different technologies of UCAR-T have been developed. The possibility of using cells from healthy donors presents several advantages such as the reduction of costs and the creation of batches, which allows the possibility of having a product immediately available also for redosing [75,76]. However, allogeneic approaches are associated with two major issues: the life-threatening GvHD and the short persistence of allogeneic CAR-T due to host immune recognition. Gene editing technology has been used to limit these drawbacks [77]. Due to the αβ TCR’s role in determining T cell alloreactivity, gene editing technology has been employed to prevent the expression of a functional TCR. A method has been developed to disrupt the gene encoding the TCR α-chain (TRAC) using Transcription Activator-Like Effector Nuclease (TALEN^®^) technology [78]. TALENs are hybrid molecules consisting of DNA recognition proteins linked to an endonuclease, which can be customized to target specific DNA sequences. The same TALEN technology has been utilized to disrupt the CD52 gene. This allows for the administration of an anti-CD52 antibody (ALLO-647) to suppress all CD52-positive immune cells capable of mediating rejection, such as T, B, and NK cells, while sparing UCART cells. In a phase 1 trial, Sallman and colleagues investigated UCART123v1.2, an allogeneic CAR-T cell product targeting CD123, in adult patients with relapsed or refractory AML. The study’s findings were promising, indicating that UCART123 combined with fludarabine, cyclophosphamide (FC), and alemtuzumab showed significantly higher UCART123 cell expansion compared to FC alone. These results support the potential use of UCART123 in the treatment of AML [79].

A further effort used the UCAR-T platform to develop novel constructs. As mentioned above, UniCAR is composed of two parts: an effector module (EM), which is the same for all or “universal”, and a target module (TM), which is specific for an antigen against tumor cells. An EM is composed of intracellular signaling domains and an extracellular site with a peptide epitope-binding domain. The TM is absent on the cell surface, so UniCAR cells had no cytotoxic function until the administration of TM. This mechanism avoids off-target effects because it blocks CAR-T cell activity by withdrawing TMs [50]. One of the most important advantages of UniCAR cells compared to CAR cells is the possibility of targeting different tumor antigens using the same UniCAR cells, modifying the TM. Additionally, UniCAR cells exhibit a good safety profile. Although the efficacy results from preclinical experiments encourage clinical application, subsequent clinical trials have not confirmed these results. Wang et al. reported an unsuccessful treatment in a relapse/refractory patient without life-threatening complications [50]. Indeed, Tambaro et al. did not find cytotoxicity against CD33 CAR-T in a phase 1 trial [43]. The next step in UniCAR technology was the development of the RevCAR platform, characterized by the small size of the gene encoding the CAR construct due to a different position of the peptide epitope and the peptide epitope-binding domain [53,80]. Furthermore, to reduce the activity of UniCAR and RevCAR cells, split, universal, and programmable (SUPRA) CAR system cells have been ideated. The SUPRA CAR system comprises two parts: leucine zipper adapter (zipCAR), which is a universal receptor, and a leucine zipper adapter (zipFv), which is single-chain variable fragment (scFv) targeting a specific antigen [81]. The scFv element presents an A leucine zipper (AZip) which can link the corresponding leucine zipper (BZip) on the zipCAR. When the zipFv binds to the target antigen, a dimer is formed, which leads to the activation of the SUPRA CAR cell. Indeed, SUPRA CAR cells present an attractive safety profile because their functions can be mitigated by changing the configuration of the leucine block. UniCAR, RevCAR, and SUPRA CAR represent novel platform technology able to reduce production costs, because only the antigen against which the TM or zipFv is directed will need to be modified.

## 9. CAR-Natural Killers Cells

CAR-NK cells have emerged as a compelling tool for immunotherapy in AML due to their ability to target and kill hematopoietic tumor cells. They possess unique properties, such as secreting interleukins, notably IFN-γ, which contribute to their effectiveness in eliminating AML cells [53]. One advantage of CAR-NK therapy is its lack of GvHD induction, as CAR-NK cells recognize the presence of HLA-I molecules on cell surfaces, sparing normal cells from cytotoxicity [82]. Additionally, CAR-NK therapy typically induces less CRS and ICANS than CAR-T therapy [81]. However, CAR-NK cells have a relatively short lifespan, necessitating multiple doses of treatment to achieve sustained CR. Moreover, CAR-NK manufacturing is more complex than CAR-T cell expansion. Consequently, Albinger et al. experimented with CD33-CAR-NK transduced with lentiviral vectors. Transduction rates of 30–60% resulted, but CD33-CAR-NK efficiently eliminated primary AML cells in vitro without CRS or GvHD [83]. The major limitation of CAR-NK therapy is the low transduction rates achieved with viral and non-viral methods, because NK cells are very sensitive to foreign DNA delivery, and in consequence CAR expression is low. To overcome this limitation, Kararoudi et al. developed a method using CRISPR/Cas9 and adeno-associated vectors (AAVs) by electroporation for specific integration on NK cells [84]. This approach resulted in improvements in cytotoxicity and cytokine secretion compared to non-modified NK cells.

## 10. Conclusions

ACT appears to be a promising strategy in the context of high-risk AML, especially for patients who have relapsed after ASCT, for whom there are very few therapeutic strategies and a worse prognosis. Currently, DLIs are effective in the context of positive MRD after ASCT, but their future use will be the maintenance post-ASCT. With regard to CAR-T, despite some encouraging outcomes from preclinical and clinical studies, significant challenges remain to be addressed before a wide clinical application. The presence of CAR-T target antigens on normal cells necessitates either the use of an anti-myeloablative strategy or CAR-T cells specifically designed to spare HSC. Furthermore, due to potential adverse events, it is crucial that CAR-T cell therapy incorporates a safety switch. Moreover, enhancing the persistence of CAR-T cells in the body and improving their functionality within the tumor microenvironment (TME) are essential to increase their efficacy. Additionally, the production of CAR-T cells may be unfeasible for some patients due to aphaeresis failure, and the lengthy manufacturing process can result in some patients succumbing to the disease before CAR-T cell administration. Therefore, ongoing research aimed at developing universal allogeneic CAR-T cells is vital. Success in this area could reduce the preparation time for CAR-T cells and lower production costs, thereby making the therapy accessible to a broader patient population.

## Figures and Tables

**Figure 1 biomedicines-12-01721-f001:**
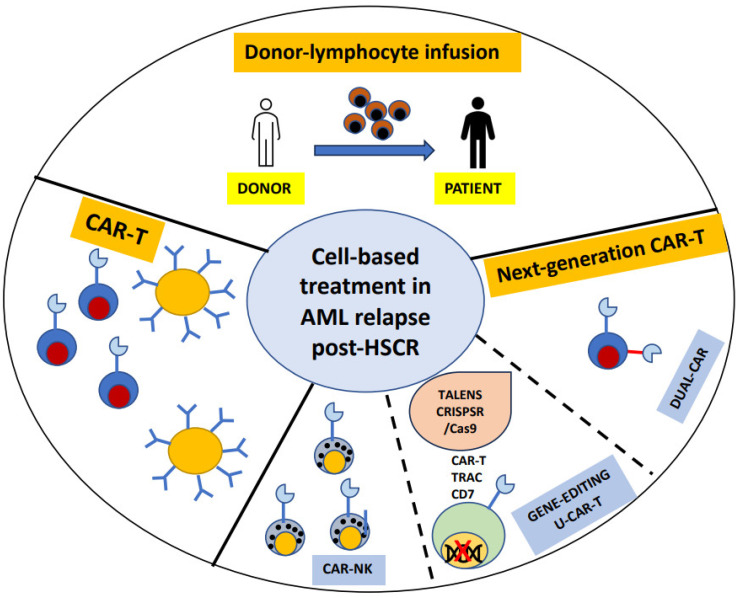
Different cellular approaches in AML relapse post-ASCT.

**Figure 2 biomedicines-12-01721-f002:**
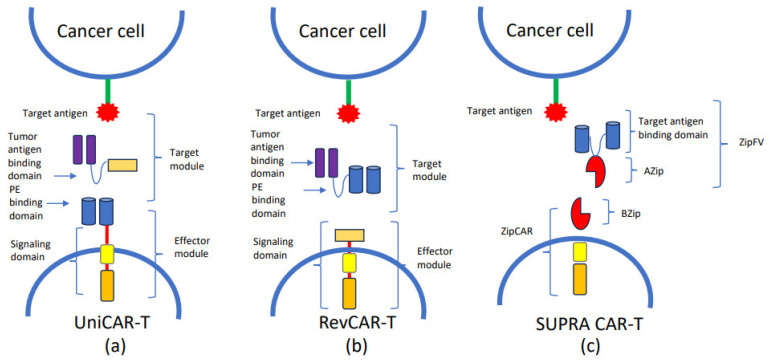
UniCAR, RevCAR and SUPRA CAR: (**a**) UniCAR is composed of a universal effector module (EM) and a tumor-specific target module (TM). EM consists of a signaling domain and a peptide epitope-binding domain, while TM comprises a peptide epitope (PE) and a tunor antigen-binding domain. The administration of TMs targets the UniCAR cell to the tumor cell, and enables the activation of cytotoxic mechanism. (**b**) RevCAR consists of universal EM and a tumor-specific target module (RevTM). EM is composed of a signaling domain and a PE. TM is a bispecific target module composed of 2 scFvs: a peptide epitope-binding domain and a tumor antigen binding domain. The administration of RevTM targets the RevCAR cell to the tumor cell, activating the cytotoxic mechanism. (**c**) SUPRA CAR is composed of a universal receptor with leucine zipper adapter (zipCAR), and scFv with a leucine zipp adapter molecule, directed against the target antigen. A leucine zipper (AZip), linked to scFv, can link to the cognate a leucine zipper (BZip) present on the zipCAR. The zipFv binding to the target antigen and dimerizing with the zipCAR results in the activation of the SUPRA CAR cell. The administration of the zipFv targets the SUPRA CAR cell to the tumor cell resulting in the activation of the cytotoxic mechanism.

**Table 1 biomedicines-12-01721-t001:** Ongoing clinical trials testing the use of anti-CD33, -CD123, -FLT3, or -CLL-1 CAR-T cells in acute myeloid leukemia therapy.

Drug	ClinicalTrials.gov Identifier	Phase of Clinical Study	Available Online
Anti-CD33 CAR-T cells	NCT04835519	Phase 1, Phase 2	https://clinicaltrials.gov/ct2/show/NCT04835519, accessed on 30 April 2023
SC-DARIC33 (anti-CD33 CAR-T cells)	NCT05105152	Phase 1	https://clinicaltrials.gov/ct2/show/NCT05105152, accessed on 30 April 2023
Anti-CD123 CAR-T cells	NCT04318678	Phase 1	https://clinicaltrials.gov/ct2/show/NCT04318678, accessed on 30 April 2023
Anti-CD123 CAR-T cells	NCT04272125	Phase 1, Phase 2	https://clinicaltrials.gov/ct2/show/NCT04272125, accessed on 30 April 2023
UCART123v1.2 (Allogeneic Engineered T cells Expressing Anti-CD123 CAR)	NCT03190278	Phase 1	https://clinicaltrials.gov/ct2/show/NCT03190278, accessed on 30 April 2023
Anti-FLT3 CAR-T	NCT05023707	Phase 1, Phase 2	https://clinicaltrials.gov/ct2/show/NCT05023707, accessed on 30 April 2023
TAA05 (anti-FLT3 CAR-T cells)	NCT05432401	Early Phase 1	https://clinicaltrials.gov/ct2/show/NCT05432401, accessed on 30 April 2023
Anti-CLL-1 CAR-T cells	NCT05252572	Early Phase 1	https://clinicaltrials.gov/ct2/show/NCT05252572, accessed on 30 April 2023
Anti-CLL-1 CAR-T cells	NCT04219163	Phase 1	https://clinicaltrials.gov/ct2/show/NCT04219163, accessed on 30 April 2023
CLL-1, CD33 and/or CD123-specific CAR-T cells	NCT04010877	Phase 1, Phase 2	https://clinicaltrials.gov/ct2/show/NCT04010877, accessed on 30 April 2023
Dual CD33/CLL-1 CAR-T	NCT05248685	Phase 1	https://clinicaltrials.gov/ct2/show/NCT05248685, accessed on 30 April 2023

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
