# Peer review of "Cell-Based Treatment in Acute Myeloid Leukemia Relapsed after Allogeneic Stem Cell Transplantation"

_biomedicines, 2024, doi:10.3390/biomedicines12081721_

Round 1

Reviewer 1 Report

Comments and Suggestions for Authors

The manuscript is well written and shows effort taken to gather and systematize the information. 

As a review there, and not a research paper, there is little to be criticized /edited.

A few issues:

Are there any data on patient outcomes /response rates in already attempted infusions in humans of non-DLI approaches?

What are /could be the briding therapies before cellular therapies in AML?

What is known about toxicity /side effects spectrum of the non-DLI therapies?

Comments on the Quality of English Language

No major issues.

Author Response

Are there any data on patient outcomes /response rates in already attempted infusions in humans of non-DLI approaches?

Thanks for this clarification. In the literature, different results of phase 1-2 clinical trials in the form of manuscript, abstracts or oral communications have been published. However, the number of the patients are too small and the follow-up is too short to lead some conclusions and to report as clinical studies. For these reasons we did not report them. Hopefully, in the next future, the results of the ongoing clinical trials together with a longer follow-up will refine the effectiveness of this treatment.

What are /could be the briding therapies before cellular therapies in AML?

The rationale of the cellular strategy based on chimeric antigen receptor (autologous or allogeneic) is the presence of an active disease. Before CAR-T administration, patients are submitter to a lymphodepleting chemotherapy based on fludarabine and Cy. In our opinion, the bridge therapy before CAR-T administration could be based on different regimen based on patient’s past history and the experience of the center. In the literature, few data have been published on this aspect. Efficacy regimen without high toxicity may be hypomethylating agents with or without venetoclax or low dose of chemotherapy. In cases of target molecular lesions, a possibility could be the specific inhibitor (Flt3, IDH1-2 inhibitors).

What is known about toxicity /side effects spectrum of the non-DLI therapies?

We added the main side effects of CAR-Ts in the paragraph 3. However, an extensive discussion of these effects is beyond the scope of this manuscript.  In each individual section we have repeatedly highlighted the problem of myelosuppression, a typical toxicity of CAR-T in AML due to the sharing of the same antigen between hematopoietic stem cells and blast cells.

Reviewer 2 Report

Comments and Suggestions for Authors

Dear Editor-in-Chief

Thank you for inviting me to review the manuscript entitled "CELL-BASED TREATMENT IN ACUTE MYELOID LEUKEMIA RELAPSED AFTER ALLOGENEIC STEM CELL TRANSPLANTATION". In the present study, the authors interestingly illustrated the therapeutic potential of ACT in relapsed AML. They discussed DIL and CAR-T cells and their clinical benefits. Although this study addressed an interesting concept, some issues should be considered before accepting the manuscript for publication:

1. In the Abstract, HSCT is closer to “hematopoietic stem cell transplantation”. I recommend changing “Allogeneic stem cell transplantation”.

2. In the Abstract, you should mention adoptive cell therapy (ACT) instead of adoptive immunotherapy. It seems that you suddenly jumped from ACT to DLI and CAR-T cell. Try to add some data to the Abstract and improve its structure, particularly in the last sentences.

3. In the Introduction, “DLI have been proven” should be changed to “DLI has been proven”.

4. The content of “2. DONOR LYMPHOCYTE INFUSIONS (DLI)” is confusing for the readers. Try separating parts of it and generating paragraphs.

5. What is the association between section “2.1. BEYOND UNMANIPULATED DLI” and 2.2 and 2.3 and …? Couldn’t you consider the following sections as subheadings of section 2.1? Furthermore, you need to add more data to those sections; they are incomplete.

6. In “2.5. DLI IN COMBINATION WITH OTHER DRUGS”, you did not mention what is AZA.

7. Try to discuss the generations of CAR-T cells, their differences, and advantages in “3. CAR-T”.

8. In the Conclusion, the authors focused on CAR-T cell therapy majorly; however, other contents of the study should be considered. Furthermore, mention the future prospects of ACT and other novel therapies in the Conclusion section.

9. There are grammatical mistakes as well as typing and spacing errors. I recommend re-checking the whole manuscript again.

Comments on the Quality of English Language

There are grammatical mistakes as well as typing and spacing errors. I recommend re-checking the whole manuscript again.

Author Response

1.In the Abstract, HSCT is closer to “hematopoietic stem cell transplantation”. I recommend changing “Allogeneic stem cell transplantation”.

Thank you, we agree with your advice and we have modified it in the main text.

  1. In the Abstract, you should mention adoptive cell therapy (ACT) instead of adoptive immunotherapy.

It seems that you suddenly jumped from ACT to DLI and CAR-T cell. Try to add some data to the Abstract and improve its structure, particularly in the last sentences.

We agree with this suggestion. We explain the ACT in the line from 13 to 15 in the abstract

  1. In the Introduction, “DLI have been proven” should be changed to “DLI has been proven”.

Thanks, we have corrected.

  1. The content of “2. DONOR LYMPHOCYTE INFUSIONS (DLI)” is confusing for the readers. Try separating parts of it and generating paragraphs.

We agree. We have divided the section into 3 paragraphs to facilitate the reader. The paragraphs focus on the three scenarios of application of DLIs which are:

- the historical approach in frank relapse.

- the current strategy when the disease is present at MRD level or with reduction of chimerism.

- the future approach which will be the use of DLI as maintenance (in prophylaxis) for high-risk.

  1. What is the association between section “2.1. BEYOND UNMANIPULATED DLI” and 2.2 and 2.3 and …? Couldn’t you consider the following sections as subheadings of section 2.1? Furthermore, you need to add more data to those sections; they are incomplete.

Thanks for this suggestion. We have modified the structure of the section 2.1 removing the paragraph 2.2 2.3 and 2.4. The rationale of this paragraph is to offer an overview regarding the different manipulation strategies of DLI though the clinical application still remains in the setting of clinical trials.

  1. In “2.5. DLI IN COMBINATION WITH OTHER DRUGS”, you did not mention what is AZA.

Thanks, we have corrected.

  1. Try to discuss the generations of CAR-T cells, their differences, and advantages in “3. CAR-T”.

Thanks, we added CAR-T generations in the main text.

  1. In the Conclusion, the authors focused on CAR-T cell therapy majorly; however, other contents of the study should be considered. Furthermore, mention the future prospects of ACT and other novel therapies in the Conclusion section.

We expanded the conclusions in the main text.

  1. There are grammatical mistakes as well as typing and spacing errors. I recommend re-checking the whole manuscript again.

We have revised the manuscript and apologize for the mistakes. We are ready, however, to make the manuscript be revised by a native English speaker if the Editor judge the corrections unsatisfactory.

Round 2

Reviewer 2 Report

Comments and Suggestions for Authors

Dear editor-in-chief,

The authors addressed my concerns satisfactorily; therefore, I recommend the manuscript be accepted in the present form.